# PRESTO: Preimage-Informed Instruction Optimization for Prompting Black-Box LLMs

**Jaewon Chu**[1]    **Seunghun Lee**[2*]  **Hyunwoo J. Kim**[2†]
[1]Korea University, [2]KAIST
allonsy07@korea.ac.kr {llsshh319, hyunwoojkim}@kaist.ac.kr

## Abstract

Large language models (LLMs) have achieved remarkable success across diverse domains, due to their strong instruction-following capabilities. This has led to increasing interest in optimizing instructions for black-box LLMs, whose internal parameters are inaccessible but widely used due to their strong performance. To optimize instructions for black-box LLMs, recent methods employ white-box LLMs to generate candidate instructions from optimized soft prompts. However, white-box LLMs often map different soft prompts to the same instruction, leading to redundant queries. While previous studies regarded this many-to-one mapping as a structure that hinders optimization efficiency, we reinterpret it as a useful prior knowledge that can accelerate the optimization. To this end, we introduce **PRE**image-informed in**ST**ruction **O**ptimization (PRESTO), a novel framework that leverages the preimage structure of soft prompts for efficient optimization. PRESTO consists of three key components: (1) score sharing, which shares the evaluation score with all soft prompts in a preimage; (2) preimage-based initialization, which selects initial data points that maximize search space coverage using preimage information; and (3) score consistency regularization, which enforces prediction consistency within each preimage. By leveraging preimages, PRESTO achieves the effect of effectively obtaining 14 times more scored data under the same query budget, resulting in more efficient optimization. Experimental results on 33 instruction optimization tasks demonstrate the superior performance of PRESTO. Code is available at https://github.com/mlvlab/PRESTO.

## 1  Introduction

Large language models (LLMs) have demonstrated strong performance across a wide range of domains [1–5]. This success is largely attributed to their impressive instruction-following capabilities, which have led to growing interest in discovering effective instructions to enhance their performance [6, 7]. In particular, LLMs provided through APIs (*i.e.,* black-box LLMs), such as GPT-4 [2], are widely used and show exceptionally strong performance. However, optimizing instructions for the black-box LLMs is a challenging problem, since their internal parameters are inaccessible. To tackle this challenge, recent studies have explored various strategies for optimizing instructions for black-box LLMs, without access to internal model parameters [8–13].

Recently, some studies [14–16] have leveraged open-source LLMs (*i.e.,* white-box LLMs) [1, 17, 18] to assist instruction optimization for black-box LLMs, demonstrating promising results and attracting growing interest. Specifically, these methods optimize a soft prompt, which is taken as input to the white-box LLM. The optimization is performed using black-box optimization algorithms such as Bayesian Optimization [19, 20] or Neural Bandits [21, 22], guided by a score predictor

---

*Work done while at Korea University.

†Corresponding author

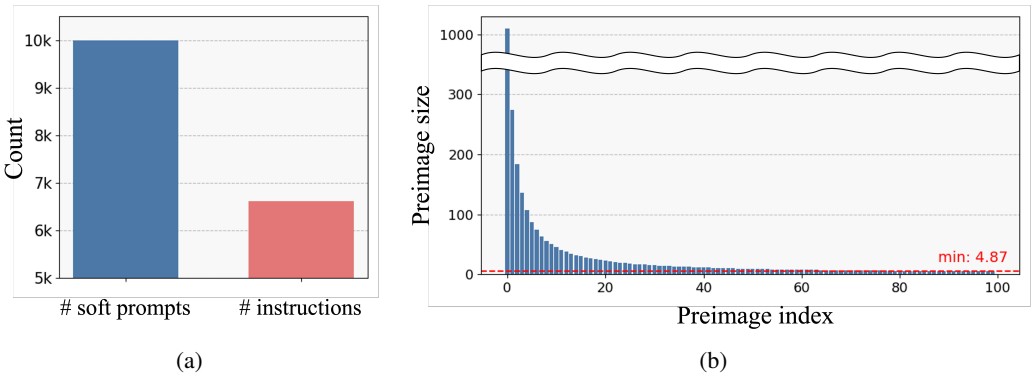

(a)                                             (b)

Figure 1: Motivating observations illustrating the many-to-one mapping from soft prompts to instructions in a white-box LLM (LLaMA3.1-8B-Instruct [1]). Figure 1a shows that the white-box LLM produces approximately 6,500 unique instructions from 10,000 distinct soft prompts. Figure 1b presents the distribution of preimage sizes, displaying the top 100 largest preimages. The largest preimage contains more than 1,000 soft prompts, while the 100th largest has around 5. Both figures report the average experimental results over the instruction induction tasks used in Table 1.

regression model, allowing the white-box LLM to generate effective instructions for black-box LLMs. However, as shown in Figure 1a, white-box LLMs often generate identical instructions from distinct soft prompts. It leads to repeatedly querying soft prompts that yield the same outputs during the optimization process, which ultimately hinders the optimization process by reducing query efficiency. To avoid redundant queries, previous studies either sample soft prompts that are well-separated in the soft prompt space [16] or filter soft prompts that generate distinct instructions [15].

While previous studies have treated the generation of identical instructions from different soft prompts (*i.e.,* many-to-one structure) as a redundancy that hinders optimization, we reinterpret this as a valuable structure that can facilitate the optimization process. Specifically, the set of soft prompts that generate the same instruction forms the preimage of that instruction under the white-box LLM. This preimage imposes a strong inductive bias over the search space: all soft prompts within a preimage share the same objective function value. Since we follow previous settings [16, 15] that sample a sufficiently large set of $N$ soft prompts and search for the optimal solution within them, we do not observe the full preimage, but only a subset of it. We refer to such subsets as preimages throughout the paper, and provide the size distribution of these preimages in Figure 1b.

Building on this insight, we propose PRESTO, a novel instruction optimization framework that explicitly leverages the many-to-one structure to facilitate instruction optimization for black-box LLMs. PRESTO consists of three components. First, we present the score-sharing method, where once the score is evaluated through the black-box LLM, it is shared with all soft prompts within a preimage. This effectively enlarges the amount of scored data without additional calls to the black-box LLM. Second, we introduce preimage-based initialization, where we select the initial soft prompts regarding the preimage information so that they cover the search space maximally. Finally, we propose score consistency regularization, which adds a regularization term to encourage the score predictor to predict identical scores for soft prompts within the same preimage. We evaluate the instruction optimization performance of PRESTO on 30 instruction induction tasks and three arithmetic reasoning tasks, and achieve state-of-the-art performance compared to existing baselines.

The main contributions of our work are:

- We reinterpret the many-to-one structure between the soft prompts and instruction, previously viewed as a challenge, as a rich informative structure that facilitates instruction optimization for black-box LLMs.

- Leveraging this insight, we introduce PRESTO, a novel framework that consists of score sharing, preimage-based initialization, and score consistency regularization.

- PRESTO achieves state-of-the-art performance across 30 instruction induction and 3 arithmetic reasoning tasks.

## 2 Related Works

**Instruction Optimization for Black-box LLMs** Instruction optimization has been widely explored as a way to improve the performance of large language models (LLMs) on downstream tasks [23, 24]. In particular, when using black-box LLMs such as GPT-4 [2], where access to model parameters is restricted, optimization methods rely on model outputs to guide the search for better instructions. Under this setting, various approaches have been proposed, including evolutionary algorithms [10, 11], LLM-driven meta-optimization [8, 9], and bandit-style or heuristic search methods [13, 12]. These works demonstrate that instruction quality can be improved even without access to gradients or internal representations by querying the black-box model efficiently.

More recently, some methods [14–16] incorporate open-source white-box LLMs [1, 18, 17, 25] to assist the optimization process. Rather than optimizing instruction texts directly, they optimize soft prompts, which are continuous embeddings that the white-box model maps into instructions. InstructZero [14] leveraged Bayesian Optimization [26–28] to search for the optimal soft prompts for black-box LLM. INSTINCT [16] leveraged NeuralUCB [21] with an LLM-based score predictor, which was the first to point out the many-to-one schema and approached it indirectly by sampling soft prompts to be well-separated. And ZOPO [15] proposed a zeroth-order optimization algorithm [29] for local search, which addresses this redundancy by simply discarding all but one soft prompt that produces the same instruction. In contrast, we retain all soft prompts by introducing preimages and facilitate the optimization.

## 3 Preliminaries

**Problem Formulation** Instruction optimization aims to find an instruction $v$ that guides a language model to perform a given task effectively. To be specific, the goal is to find the instruction $v$ that maximizes the task-specific score function $h$ by guiding a black-box LLM $f_b$ to generate the correct answer $y$, which is formally given as:

$$v^* = \arg\max_{v \in \Omega} \mathbb{E}_{(x,y) \in D_{\text{val}}} \big[ h(f_b(v, x), y) \big], \tag{1}$$

where $\mathcal{D}_{\text{val}} = \{(x_i, y_i)\}_{i=1}^M$ is a validation set, and $\Omega$ denotes the search space of instructions, typically a discrete sequence domain (*e.g.,* natural language prompts or token sequences). However, directly searching over discrete instruction sequences is challenging, as it constitutes a combinatorial optimization problem over the space of token configurations. To address this, InstructZero [14] reformulates the discrete instruction search as a continuous optimization problem by leveraging a white-box LLM $f_w$. Specifically, it optimizes a soft prompt $z \in \mathbb{R}^{N_z \times d}$, where $N_z$ is the number of tokens and $d$ is the embedding dimension, to generate the optimal instruction $v^*$. The soft prompt is concatenated with the token embeddings of input-output exemplars $E = \{(x_i, y_i)\}_{i=1}^\kappa$ and fed into the white-box LLM $f_w$, which then generates an instruction $v = f_w(z, E)$. Formally, the instruction optimization problem is defined as:

$$z^* = \arg\max_{z \in \mathcal{Z}} \mathbb{E}_{(x,y) \in D_{\text{val}}} \big[ h(f_b(f_w(z, E), x), y) \big], \tag{2}$$

where $\mathcal{Z}$ is the soft prompt space. In this formulation, we optimize $z$ to find the optimal instruction $v^*$ that maximizes the expected value of the score function $h$. Once the optimal soft prompt $z^*$ is obtained, the corresponding instruction $v^*$ is generated by the white-box LLM $f_w$, *i.e.*, $v^* = f_w(z^*, E)$ and subsequently evaluated on a held-out test set $D_{\text{test}}$. Since the exemplars $E$ are fixed for each task, we omit them from the notation in the rest of our paper. Following previous works [14–16], we assume that both the white-box LLM $f_w$ and the black-box LLM $f_b$ are deterministic.

**LLM-based Score Predictor for Instruction Optimization.** Our method builds upon IN-STINCT [16], which employs a frozen white-box LLM as a feature extractor to predict the score of soft prompt, and uses a NeuralUCB [21] for instruction optimization. Given a soft prompt $z$, the white-box LLM produces an embedding $g(z)$, the last token representation of the final transformer layer. This embedding is then passed to a score predictor $m(g(z); \theta)$ (*e.g.*, an MLP), which predicts the performance of the instruction generated from $z$, *i.e.*, $m(g(z); \theta) \approx \mathbb{E}_{(x,y) \in D}[h(f_b(f_w(z), x), y)]$. At each optimization step, the score predictor $m(\cdot; \theta)$ is trained on previously evaluated soft prompts and their corresponding scores, and selects the next query that maximizes the upper confidence bound. We provide further details of NeuralUCB in the supplement. Since computing $g(z)$ requires a full

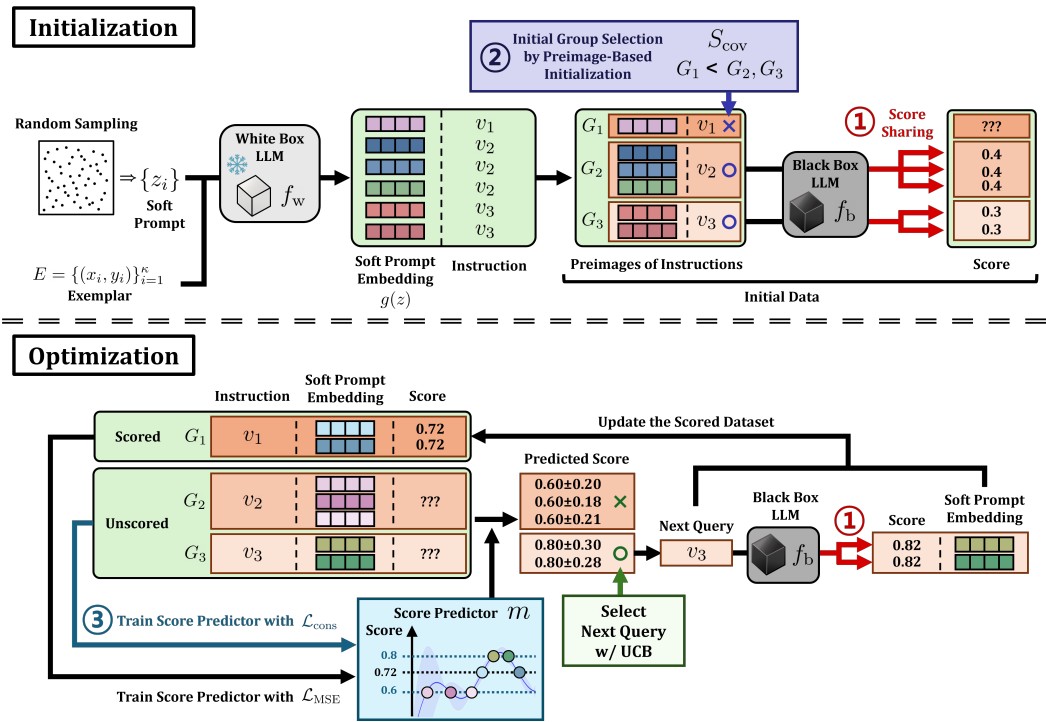

Figure 2: **The overall process of our proposed PRESTO framework**. It consists of two main stages: initialization and optimization. In the initialization stage, our method performs ① *preimage-based score sharing* (Section 4.1) and ② *preimage-based initialization* to improve search space coverage (Section 4.2). For the optimization stage, we train the score predictor with ③ *score consistency regularization* (Section 4.3) and we apply ① *preimage-based score sharing* to share scores of newly observed data within the same preimage.

forward pass through the LLM, INSTINCT mitigates this cost by precomputing the embeddings of a candidate soft prompt set $Z = \{z_i\}_{i=1}^N$ at the beginning of the optimization, which is sampled using a quasi-random method. To this end, the instruction optimization task is reduced to searching for the best solution within the precomputed embedding set, as the white-box LLM is frozen during the optimization process.

# 4   Method

In this section, we propose **PRE**image-informed in**ST**ruction **O**ptimization (PRESTO) which is a novel instruction optimization framework that leverages the many-to-one mapping between soft prompts $z \in Z \subset \mathcal{Z}$ and instructions $v \in \Omega$ (or the preimages of instructions, which is defined in Section 4.1) as prior knowledge to facilitate more efficient optimization. We first introduce a score sharing method that shares the score value of one soft prompt with all other soft prompts in the same preimage, effectively enlarging the scored data without additional evaluations of black-box LLM $f_{\mathrm{b}}$. Next, we present a preimage-based initialization method designed to maximize coverage of the search space under score sharing. Finally, we propose a score consistency regularization that leverages preimage information as prior knowledge to encourage the score predictor to predict identical scores for soft prompts belonging to the same preimage. We provide the overall framework of our PRESTO in Figure 2.

## 4.1   Preimage-Based Score Sharing

During the instruction optimization, we observe that the white-box LLM $f_{\mathrm{w}}$ often generates identical instructions from distinct soft prompts, *i.e.*, $f_{\mathrm{w}}(z) = f_{\mathrm{w}}(z')$, leading to the same score value. This redundancy leads to unnecessary queries during optimization, hindering the efficiency of instruction

optimization. While previous works treated this redundancy as an obstacle to efficient optimization, we instead leverage this information as prior knowledge about the objective function to facilitate optimization. To this end, we propose a simple score sharing scheme that associates a large number of soft prompts with a score value without the additional evaluations of a black-box LLM $f_{\mathrm{b}}$.

Our goal is to share the score of an evaluated soft prompt $z$ with other soft prompts that generate the same instruction. To enable this score sharing, we first define the *preimage* of each instruction which consists of all soft prompts that map to the same instruction under the white-box model $f_{\mathrm{w}}$. Establishing this preimage structure requires two steps. First, we sample a soft prompt set $Z = \{z_i\}_{i=1}^N$ using a quasi-random method [30, 31], which is a widely adopted method to sample the data points that evenly cover the soft prompt space [14, 16, 15]. Assuming that the soft prompt set size $N$ is large enough to represent the soft prompt space $\mathcal{Z}$, the original optimization problem defined in Eq. (2) reduces to searching for the best solution among the set of $N$ data points, denoted by $Z \subset \mathcal{Z}$.

Next, for each soft prompts $z_j \in Z$, we generate the set of instructions $V = \{v_i\}_{i=1}^M$, using the white-box LLM $f_{\mathrm{w}}$:

$$V = \{v_i\}_{i=1}^M = \{f_{\mathrm{w}}(z_j) \mid j = 1, \dots, N\}. \tag{3}$$

Since the different soft prompts often generate the identical instruction (*i.e.,* many-to-one mapping), the number of instructions $M = |V|$ is smaller than or equal to $N$. The construction of $Z$ and $V$ is performed only once before the optimization process begins.

With the soft prompt set $Z$ and the corresponding instruction set $V$, we now define the preimage of each instruction. The preimage of an instruction $v$ is the set of soft prompts in $Z$ that generate $v$ under the white-box model $f_{\mathrm{w}}$:

$$f_w^{-1}(v) = \{z \in Z \mid f_{\mathrm{w}}(z) = v\}. \tag{4}$$

This preimage contains all soft prompts in $Z$ that generate $v$, and will serve as the basis for score sharing. Once the preimages $f_{\mathrm{w}}^{-1}(v)$ for all $v \in V$ are established, we apply score sharing across soft prompts that belong to the same preimage during the optimization. Specifically, after querying the black-box model $f_{\mathrm{b}}$ with an instruction $v \in V$, we obtain a score of the instruction. This score is then shared to all soft prompts in the preimage $f_{\mathrm{w}}^{-1}(v)$. By sharing scores in this manner, we effectively enlarge the training data for the score predictor $m(g(z); \theta)$ without additional calls to the black-box LLMs. Moreover, score sharing avoids redundant evaluations of soft prompts that lead to the same instruction and improves optimization efficiency.

## 4.2 Preimage-Based Initialization for Maximizing Search Space Coverage

Here, we introduce a preimage-based initialization method that selects initial data points based on the preimage information defined in Section 4.1. At the beginning of the optimization, the score predictor $m(g(z); \theta)$ (Section 3) is trained on the initial dataset, and its predictions are used to select the next data points to query the black-box LLM $f_{\mathrm{b}}$. In black-box optimization, it is well known that broadly covering the search space at initialization is crucial for effective optimization [32–35]. Our score sharing method introduced in Section 4.1 expands the initial dataset without additional queries to the black-box LLM $f_{\mathrm{b}}$, enabling a more sample-efficient initialization. To further enhance the search space coverage, we propose a preimage-based initialization method that complements score sharing by promoting a broader initial data distribution.

To this end, we design a coverage score $S_{\mathrm{cov}}$ to guide the selection of an initial preimage set $G^{\mathrm{init}}$ that maximally covers the entire set of soft prompt embeddings $G^{\mathrm{total}} = \{g(z) \mid z \in Z\}$. We conduct initialization in the embedding space rather than the raw soft prompt space, since the optimization operates over the soft prompt embeddings. These embeddings are precomputed and remain fixed throughout the optimization, as described in Section 3. For each instruction $v_i$, we define its corresponding preimage group in the embedding space as $G_i = \{g(z) \mid z \in f_{\mathrm{w}}^{-1}(v_i)\}$.

Since finding the optimal combination of $N_{\mathrm{init}}$ preimages that maximizes the coverage score $S_{\mathrm{cov}}$ is a computationally intractable combinatorial optimization problem, we adopt a greedy algorithm to iteratively select one preimage at a time. Specifically, the coverage score $S_{\mathrm{cov}}$ consists of two components: the representativeness score $S_{\mathrm{rep}}$ and the size score $S_{\mathrm{size}}$. The representativeness score $S_{\mathrm{rep}}$ encourages the selection of a preimage group $G_i$ that, when combined with already selected

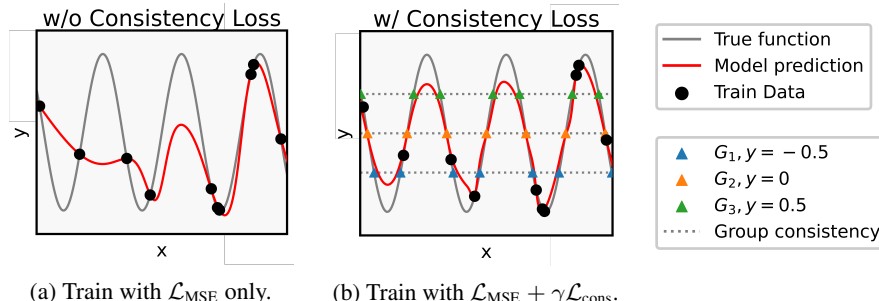

(a) Train with $\mathcal{L}_{\text{MSE}}$ only.  (b) Train with $\mathcal{L}_{\text{MSE}} + \gamma\mathcal{L}_{\text{cons}}$.

Figure 3: **Toy example** comparing models trained w/o and w/ our consistency loss $\mathcal{L}_{\text{cons}}$ in Eq. (8).

preimage groups $G^{\text{init}}$, most closely matches the distribution of the candidate set $G^{\text{total}}$, defined as:

$$S_{\text{rep}}(G_i; G^{\text{init}}, G^{\text{total}}) = 1 - \frac{\text{MMD}^2(G_i \cup G^{\text{init}}, G^{\text{total}})}{\max_j \text{MMD}^2(G_j \cup G^{\text{init}}, G^{\text{total}})}, \tag{5}$$

where the $\text{MMD}^2$ is the squared Maximum Mean Discrepancy. $\text{MMD}^2$ is a widely used metric to estimate the similarity between two sets, which is defined as:

$$\text{MMD}^2(X, Y) = \mathbb{E}_{x,x'\sim X}[k(x, x')] + \mathbb{E}_{y,y'\sim Y}[k(y, y')] - 2\mathbb{E}_{x\sim X, y\sim Y}[k(x, y)] \tag{6}$$

where $k(\cdot, \cdot)$ is a positive definite kernel. To densely cover the search space, we propose the size score $S_{\text{size}}$, which is defined as relative preimage size: $S_{\text{size}}(G_i) = |G_i|/\max_j |G_j|$. Combining the two scores, we define the coverage score for the $G_i$:

$$S_{\text{cov}}(G_i; G^{\text{init}}, G^{\text{total}}) = S_{\text{size}}(G_i) + S_{\text{rep}}(G_i; G^{\text{init}}, G^{\text{total}}). \tag{7}$$

Starting from an empty set $G^{\text{init}}$, we iteratively select the preimage with the highest coverage score $S_{\text{cov}}$ and add it to $G^{\text{init}}$ until the number of initial preimages reaches $N_{\text{init}}$. This initialization maximizes the coverage of the candidate set $G^{\text{total}}$. We provide the visualization to demonstrate the effectiveness of our initialization method in Section 6.3.

### 4.3 Score consistency regularization for score predictor

Here, we propose a score consistency regularization that encourages the score predictor $m(g(z); \theta)$ to produce the same prediction for all soft prompts in preimages that have not been evaluated by the black-box function. During the optimization, the score predictor is trained with the scored data to predict the score of each soft prompt in the candidate set $Z$ and estimate its uncertainty for selecting the next query to evaluate. Leveraging the score sharing method defined in Section 4.1 informs the score predictor that data points within the same preimage share identical scores in a supervised manner. However, since the score predictor lacks information about score consistency within unscored preimages, it is unable to make consistent predictions for data points in these unscored preimages. It often hinders the score predictor from predicting the ground truth score and selecting high-scored data.

To ensure consistent predictions within each unscored preimage, we propose a score consistency regularization term $\mathcal{L}_{\text{cons}}$, which is defined as:

$$\mathcal{L}_{\text{cons}} = \mathbb{E}_{v\in V_{\text{unseen}}}\mathbb{E}_{z,z'\in f_{\text{w}}^{-1}(v)} |m(g(z); \theta) - m(g(z'); \theta)|^2, \tag{8}$$

where $V_{\text{unseen}} \subset V$ denotes the set of instructions that has not been evaluated by the black-box LLM $f_b$. We note that $\mathcal{L}_{\text{cons}}$ is an unsupervised loss. While the consistency regularization includes pairwise terms per preimage group, each unscored preimage size is not excessively large in practice, so the computation remains tractable. The final loss for training the score predictor model is given by:

$$\mathcal{L} = \mathcal{L}_{\text{MSE}} + \gamma\mathcal{L}_{\text{cons}}, \tag{9}$$

where $\mathcal{L}_{\text{MSE}}$ is the mean squared error loss computed over the scored preimages, and $\gamma$ is a hyperparameter controlling the strength of the regularization. To avoid premature convergence to incorrect predictions, we employ a simple linear scheduling strategy as $\gamma(t) = \gamma_{\text{max}} \cdot \min(1, t/T)$, where $t$

Table 1: Performance on instruction induction tasks. Bolded numbers (blue) indicate the best methods for each task. Scores show the average accuracy with standard error over three runs.

| Tasks | APE | InstructZero | INSTINCT | EvoPrompt | ZOPO | OPRO ‖ | PRESTO |
|---|---|---|---|---|---|---|---|
| antonyms | 80.67 ± 0.72 | 75.33 ± 3.21 | **83.33** ± 0.54 | 82.00 ± 0.47 | 82.67 ± 1.66 | 80.33 ± 2.33 | **83.33** ± 1.19 |
| auto_categorization | 26.00 ± 6.13 | 27.67 ± 2.60 | 18.67 ± 0.72 | 29.33 ± 2.18 | **31.67** ± 3.41 | 30.33 ± 0.72 | **31.67** ± 3.41 |
| auto_debugging | 8.33 ± 6.80 | 12.50 ± 5.89 | 10.00 ± 4.71 | 16.67 ± 6.80 | 13.33 ± 7.20 | 8.33 ± 6.80 | **20.83** ± 3.40 |
| cause_and_effect | 92.00 ± 1.89 | 74.67 ± 4.75 | 76.00 ± 9.98 | 72.00 ± 6.80 | 93.33 ± 2.88 | 38.67 ± 4.35 | **94.67** ± 2.88 |
| common_concept | 22.36 ± 2.34 | 15.53 ± 5.11 | 20.21 ± 1.19 | 17.99 ± 6.72 | 21.86 ± 7.16 | 20.08 ± 6.70 | **22.86** ± 3.27 |
| diff | 18.33 ± 6.87 | 53.00 ± 20.37 | 81.67 ± 13.76 | 7.00 ± 5.72 | 88.33 ± 5.93 | 64.33 ± 23.91 | **98.00** ± 0.82 |
| informal_to_formal | 57.59 ± 2.40 | 51.53 ± 4.62 | 48.93 ± 3.46 | 42.87 ± 2.03 | **58.93** ± 4.83 | 50.02 ± 2.63 | 52.77 ± 5.46 |
| letters_list | 99.00 ± 0.82 | 99.00 ± 0.47 | 97.67 ± 1.52 | 73.67 ± 9.69 | 98.67 ± 1.09 | 99.00 ± 0.47 | **99.33** ± 0.54 |
| negation | 83.33 ± 1.19 | 81.67 ± 3.95 | 76.67 ± 4.77 | 71.67 ± 1.19 | 77.33 ± 4.63 | 73.33 ± 4.23 | **84.00** ± 2.16 |
| object_counting | 37.33 ± 5.50 | 46.00 ± 5.72 | **48.67** ± 3.21 | 28.67 ± 2.23 | 34.00 ± 4.08 | 31.00 ± 3.86 | 45.67 ± 4.38 |
| odd_one_out | 51.33 ± 14.43 | 46.67 ± 5.76 | 60.00 ± 7.12 | 68.00 ± 1.89 | 58.67 ± 7.14 | 47.33 ± 10.39 | **70.00** ± 0.94 |
| orthography_starts_with | 46.00 ± 8.18 | 35.00 ± 3.56 | 54.67 ± 8.20 | 42.00 ± 15.28 | 54.67 ± 3.66 | 22.33 ± 10.18 | **57.33** ± 6.08 |
| rhymes | 69.33 ± 16.41 | 81.67 ± 10.69 | **98.67** ± 0.72 | 93.67 ± 1.96 | 83.33 ± 6.87 | 77.00 ± 15.25 | 85.00 ± 7.41 |
| second_word_letter | 72.67 ± 10.88 | 40.67 ± 5.99 | 48.00 ± 22.38 | 33.00 ± 7.93 | 68.00 ± 17.75 | 22.00 ± 14.73 | **77.00** ± 12.57 |
| sentence_similarity | **29.00** ± 5.44 | 17.33 ± 4.75 | 11.33 ± 5.42 | **29.00** ± 0.47 | 4.33 ± 3.54 | 6.67 ± 5.44 | 21.67 ± 8.49 |
| sum | 24.00 ± 14.61 | 55.00 ± 23.92 | 99.33 ± 0.54 | 66.67 ± 27.22 | **100.00** ± 0.00 | 91.33 ± 3.78 | 94.67 ± 4.35 |
| synonyms | 10.00 ± 4.50 | 22.67 ± 5.62 | 25.00 ± 8.83 | **25.33** ± 7.98 | 24.33 ± 2.76 | 12.67 ± 0.72 | 18.33 ± 1.91 |
| taxonomy_animal | 43.67 ± 15.96 | 44.33 ± 17.72 | 92.00 ± 3.77 | 34.00 ± 24.10 | 69.00 ± 24.10 | 73.67 ± 8.09 | **99.67** ± 0.27 |
| word_sorting | 54.00 ± 15.41 | 39.67 ± 12.11 | 27.33 ± 7.37 | **71.00** ± 4.50 | 54.00 ± 15.06 | 36.33 ± 11.49 | 53.33 ± 8.38 |
| word_unscrambling | 28.00 ± 4.78 | 38.00 ± 3.74 | 42.33 ± 8.59 | 23.00 ± 9.57 | **52.00** ± 7.79 | 43.00 ± 1.25 | 48.00 ± 7.59 |
| **# best-performing tasks** | 1 | 0 | 3 | 3 | 4 | 0 | **12** |
| **Average Rank** | 4.25 | 4.80 | 3.70 | 4.70 | 3.05 | 5.20 | **1.90** |

represents the current epoch and $T$ is a warm-up duration. This schedule allows the score predictor $m(g(z); \theta)$ to learn accurate patterns from the scored data and gradually incorporate the score equality constraint of unscored data.

Figure 3 shows a toy example illustrating the effect of the proposed consistency loss. We use a simple model with two linear layers. In Figure 3a, the model is trained only with the $\mathcal{L}_{\text{MSE}}$ on the scored data, while in Figure 3b, $\mathcal{L}_{\text{cons}}$ is additionally applied to unscored data. We assume there are three unscored preimages, each represented by a different marker shape. Although the model is only given the information that data points within each preimage share the same score, the $\mathcal{L}_{\text{cons}}$ allows it to make more accurate predictions on the unscored data.

## 5 Experiments

### 5.1 Experimental settings

We evaluate our proposed method, PRESTO, on 30 instruction induction tasks [36], a benchmark widely used to assess instruction optimization performance, and 3 arithmetic reasoning tasks [37–39]. We compare PRESTO with six competitive instruction optimization baselines: APE [8], InstructZero [14], INSTINCT [16], EvoPrompt [10], ZOPO [15], and OPRO [9]. We use LLaMA3.1-8B-Instruct [1] as the white-box LLM $f_w$ to generate candidate instructions, and GPT-4.1 as the black-box model $f_b$. Following previous works [14–16], we set the total query budget to 165, initialize with 40 soft prompts, and evaluate all methods over three different random seeds. To ensure a fair comparison, we follow the hyperparameter tuning procedure in [16]. Detailed hyperparameter configurations and experimental settings are provided in the supplement.

### 5.2 Instruction induction results

Here we provide the results of our proposed method, PRESTO, compared with six strong baselines on instruction induction tasks. To enhance readability, we report results on a subset of 20 following previous works [16, 15]. The full results for all 30 tasks are provided in the appendix. Table 1 shows that PRESTO achieves the highest accuracy on 12 out of the 20 tasks, which is three times more than the second-best method, ZOPO. In addition, PRESTO attains the best average rank of 1.90, outperforming all baselines by a clear margin; the next best, ZOPO, has an average rank of 3.05, followed by INSTINCT at 3.70. These results highlight the strong performance of PRESTO on individual tasks and its robustness across a wide range of instruction induction tasks. In the full set of 30 tasks, PRESTO also consistently outperforms other baselines with a large margin in the number of best-performing tasks and average rank.

Table 2: Performance of different CoT prompts on three math reasoning datasets. The best result for each dataset is in **bold**, and the second best is underlined.

| Method | Dataset | Best instruction | Accuracy |
|---|---|---|---|
| Hand-crafted | GSM8K | Let's think step by step | 0.9121 |
| InstructZero | GSM8K | Let's think step by step to solve the math problem | 0.9083 |
| INSTINCT | GSM8K | Let's break down and solve the problem | 0.9098 |
| ZOPO | GSM8K | Let's break it down and find the solution | **0.9143** |
| PRESTO (Ours) | GSM8K | Let's break it down together | 0.9128 |
| Hand-crafted | AQUA-RAT | Let's think step by step. | 0.7402 |
| InstructZero | AQUA-RAT | Let's break it down and find the solution | 0.7480 |
| INSTINCT | AQUA-RAT | Let's break it down step by step. I am ready to solve the problem. | 0.7480 |
| ZOPO | AQUA-RAT | Let's break it down mathematically. | 0.7520 |
| PRESTO (Ours) | AQUA-RAT | Let's solve it together. | **0.7756** |
| Hand-crafted | SVAMP | Let's think step by step. | 0.9375 |
| InstructZero | SVAMP | Let's crack the code! | **0.9400** |
| INSTINCT | SVAMP | Let's break it down step by step | 0.9375 |
| ZOPO | SVAMP | I see what you're doing there | **0.9400** |
| PRESTO (Ours) | SVAMP | Let's use the formula | **0.9400** |

Table 3: Ablation study of **PRESTO**. We incrementally add score sharing (**SS**, Sec. 4.1), preimage-based initialization (**Init**, Sec. 4.2), and consistency regularization (**Reg**, Sec. 4.3) to a vanilla baseline.

| Model | SS | Reg | Init | # Wins | Avg. Rank | Avg. acc. |
|---|---|---|---|---|---|---|
| Vanilla | ✗ | ✗ | ✗ | 0 | 4.55 | 51.91 |
| + SS | ✓ | ✗ | ✗ | 3 | 3.10 | 59.57 |
| + SS + Reg | ✓ | ✓ | ✗ | 4 | 2.65 | 61.77 |
| + SS + Init | ✓ | ✗ | ✓ | 4 | 2.30 | 61.82 |
| + SS + Init + Reg (**Ours**) | ✓ | ✓ | ✓ | **9** | **2.20** | **62.91** |

## 5.3 Chain-of-Thought Prompting Results

We evaluate the quality of the optimized instructions by measuring their effectiveness as chain-of-thought (CoT) [40] prompts on three math reasoning benchmarks: GSM8K [37], AQUA-RAT [38], and SVAMP [39]. We compare our method with three baselines that use soft prompts (InstructZero [14], INSTINCT [16], and ZOPO [15]), as well as a standard hand-crafted prompt [41]. Table 2 demonstrates that our PRESTO outperforms or matches the best-performing baselines across all datasets. In particular, it achieves the highest accuracy on AQUA-RAT (0.7756) and ties for the best result on SVAMP (0.9400), while remaining competitive on GSM8K. These results indicate that the instructions optimized by our method are also effective when used as CoT prompts.

## 6 Analysis

### 6.1 Ablation Study

We perform an ablation study to analyze the contribution of each component in our method over the 20 instruction induction tasks used in Table 1 over 3 random seeds. Starting from a vanilla baseline without our techniques, we incrementally add: (1) score sharing method (Section 4.1), (2) preimage-based initialization (Section 4.2), and (3) score consistency regularization (Section 4.3). The full model with all components combined corresponds to our proposed method, PRESTO. As shown in Table 3, each component contributes to performance improvement. In particular, introducing score sharing significantly boosts accuracy from 51.91 to 59.57 (+7.66) and improves average rank from 4.55 to 3.10 (-1.45), indicating its strong impact. Our PRESTO achieves the best results overall, with the highest number of wins and the lowest average rank across tasks.

### 6.2 Impact of score sharing method

We report the average number of soft prompts with assigned scores after the optimization process, comparing our method with baselines across all 30 tasks. The reported count includes soft prompts that were scored either directly through black-box evaluation or indirectly via score sharing. As shown in Figure 4, our method assigns scores to over 2,300 soft prompts on average, 14× more than

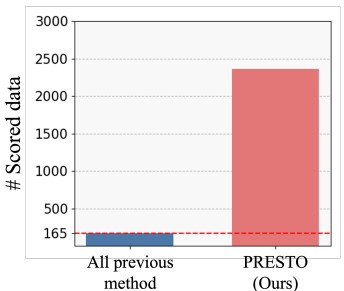

Figure 4: Average number of scored soft prompts after optimization across all tasks.

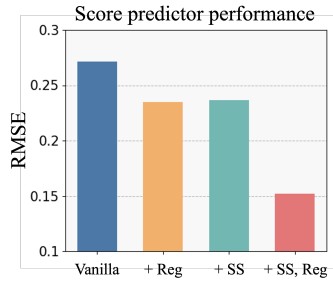

Figure 5: Performance of score predictor trained with diverse methods.

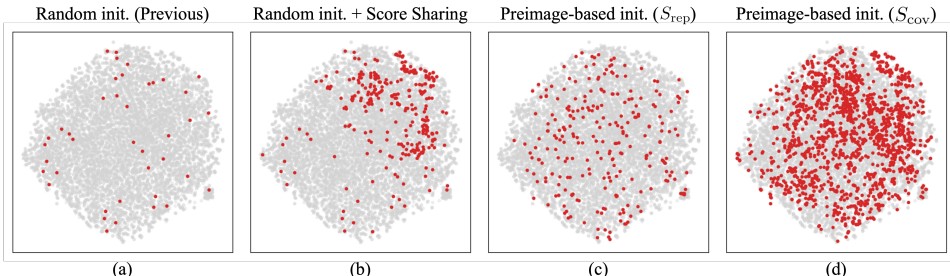

Figure 6: Visualization of the initial data distribution under different initialization. We plot the entire soft prompt embedding candidate set $G^{\text{total}}$ using t-SNE, and highlight the selected initial data in red.

previous methods, which yield only 165 scored data points, equal to the query budget in our setting. The large amount of scored data enables the score predictor to learn the objective function more effectively, which in turn facilitates more successful optimization. This analysis demonstrates that our score-sharing method can significantly increase the amount of scored data without requiring additional black-box queries.

## 6.3 Visualization of Preimage-Based Initialization

We present a qualitative analysis of how score sharing and preimage-based initialization influence the distribution of initial soft prompts. Figure 6 visualizes the distribution of initial soft prompts under four settings: (1) random initialization, (2) random initialization with score sharing (Section 4.1), (3) preimage-based initialization using $S_{\text{rep}}$ only, and (4) $S_{\text{cov}} = S_{\text{rep}} + S_{\text{size}}$ (Section 4.2) in "objective counting" task. To visualize the spatial distribution of soft prompt embeddings, we employ t-SNE. Compared to random initialization in prior works, score sharing enlarges the size of the initial dataset without additional black-box queries. Furthermore, selecting initial data using $S_{\text{rep}}$ leads to better coverage of the soft prompt space than naive score sharing. Finally, our proposed preimage-based initialization method that utilizes $S_{\text{cov}}$ achieves the densest and comprehensive coverage of the search space. It shows that our preimage-based initialization method effectively selects the initial data points that densely and evenly cover the search space.

## 6.4 Score Predictor Performance Enhancement

To analyze how score sharing (Section 4.1) and score consistency regularization (Section 4.3) influence the quality of the score predictor, we evaluate its prediction performance under different training configurations. Figure 5 reports the root mean squared error (RMSE), where lower values indicate higher prediction accuracy. We use 100 randomly selected soft prompts as training data and another 100 as test data for the objective counting task. As shown in Figure 5, applying either score sharing or score consistency regularization improves the score predictor's performance, reducing the RMSE from approximately 0.27 (vanilla) to around 0.23. When both techniques are applied together, the RMSE further decreases to approximately 0.15, indicating a strong complementary effect. The results demonstrate that expanding the training set without requiring additional black-box queries

through score sharing and incorporating the preimage structure as a prior via score consistency regularization are both crucial for enhancing the score predictor's performance.

# 7 Conclusion

We propose PRESTO, a preimage-informed instruction optimization framework that explicitly leverages this many-to-one structure via preimage. PRESTO consists of three components that leverage the preimage structure: score sharing to propagate labels within each preimage, preimage-based initialization to improve search space coverage, and consistency regularization to align predictions within unscored preimages. PRESTO achieves state-of-the-art performance on 33 instruction optimization tasks, and our comprehensive analysis supports its effectiveness and robustness.

**Limitations and broader impacts**

Our method introduces preimage-based score sharing to enlarge the number of data, which incurs mild computational overhead compared to simpler baselines. Moreover, its benefits are more pronounced when applied to a large candidate set, as score sharing is most effective when many soft prompts map to the same instruction.

In terms of broader impact, this work aims to make black-box LLM optimization more data-efficient, which can reduce the cost of experimentation and improve accessibility for researchers with limited resources. However, as with any optimization technique for LLMs, there is a risk that improved performance could be applied in ways that reinforce biases or generate harmful content. Careful deployment and alignment with responsible AI principles are necessary.

**Acknowledgement**

This research was supported by the ASTRA Project through the National Research Foundation (NRF) funded by the Ministry of Science and ICT (No. RS-2024-00439619).

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
