# OpenReview forum: "PRESTO: Preimage-Informed Instruction Optimization for Prompting Black-Box LLMs"
_NeurIPS.cc/2025/Conference — NeurIPS 2025 poster_

### Official Review · Reviewer_cs11 · 2025-07-01

**Clarity:** 3
**Significance:** 3
**Originality:** 2
**Rating:** 4
**Confidence:** 4

**Summary:**

This paper tackles inefficient instruction optimization for black-box LLMs, a problem arising when different "soft prompts" generate the same instruction. The authors reframe this many-to-one mapping as a useful "pre-image" structure and introduce PRESTO, a framework that leverages it for efficient optimization. PRESTO employs three techniques: score sharing, which propagates a single evaluated score to all soft prompts within its preimage; preimage-based initialization, which improves initial data selection to maximize search space coverage; as well as score consistency regularization, which enforces consistent predictions for prompts within the same preimage. Over dozens of different tasks, PRESTO can achieve comparably or better performance on instruction optimization tasks compared with baselines.

**Questions:**

Please kindly refer to my comments above

**Ethical Concerns:**

["NO or VERY MINOR ethics concerns only"]

**Final Justification:**

I appreciate the new wall-clock time comparison, which helps address my questions about overhead.

I still believe that the work can be enhanced with deeper theoretical justification for its heuristic design choices. Furthermore, a qualitative analysis is still needed to explain why the method's instructions are more effective on complex tasks. This would provide valuable insight that the dry accuracy metrics alone do not capture.

As mentioned in my initial review, I will maintain my positive rating given the paper's strong empirical contributions.

**Limitations:**

There is a limitations section after Section 7

**Quality:**

2

**Strengths And Weaknesses:**

Positive:

1. The manuscript is generally well-structured and easy to follow. Figure 2 also provides a clear high-level illustration and narrative of the overall pipeline. Augmenting the data to provide abundant learning signals without additionally querying the expensive API is intuitive.

2. The method's robustness is validated through extensive experiments across various combinations of white-box (LLaMA, Qwen, Mistral) and black-box (GPT-4.1, Gemini-2.0-Flash) language models. Statistical significance results are also reported via standard deviation. The extensive experiments are the main contribution of this work.

3. Code implementation is provided for reproducing the results.


Negative:

1.	The method is an extension of the existing INSTINCT framework, using the related NeuralUCB algorithm and codebase, leading to limited algorithmic contributions and limited novelty. The core techniques are based on well-established machine learning concepts, including distribution matching with MMD for initialization, and consistency regularization for training.

2.	The training pipeline still needs further justifications since it is purely heuristic rather than theoretically justified by the authors. For instance, what is the motivation and justification of designing the coverage score as Eq (7) other than simple heuristics? The toy example along is also not sufficient since it is merely a toy example with a small network of two layers as suggested by the authors.

3.	While an efficiency analysis is provided, the paper lacks an explicit wall-clock time comparison with the baselines, which can better demonstrate the computational overhead of the new components, especially with the newly added soft prompt augmentation techniques.

4.	While the experiments cover over 30 tasks, most of them are simple induction tasks. Although additional evaluations on three math reasoning tasks are included in Table 2, there are no significant insights derived from these empirical results, as the best instructions across methods are very similar and notably short. The authors also provide no further analysis or justification for the advantages of the generated instructions produced by their method, other than good performance (248-255). Given the small performance gaps and the highly similar instructions, it is difficult to justify the superiority of the proposed approach over baselines, on the other tasks beyond simple instruction induction.

---

> ### Author Rebuttal · Authors · 2025-07-31
>
> - **[W1] Contributions and novelty of PRESTO**
>
>     Our main contribution is that we reinterpret many-to-one mapping between soft prompts and instructions—typically, previous works considered it as a challenge of white-box LLM-based instruction optimization for black-LLMs—as useful prior knowledge for optimization by leveraging the preimage structure. To our knowledge, this is the first work to leverage the preimage structure in instruction optimization. Also, both reviewers uyym and BKWA highlighted the novelty as a strength of our PRESTO, stating that "The idea of reinterpreting the many-to-one mapping as useful prior knowledge is novel" and "The key insight is both original and well-motivated.", respectively.
>
> - **[W2] Justification of the coverage score and consistency regularization**
>
>     While PRESTO does not rely on a formal theoretical proof, its design is grounded in widely accepted principles in black-box optimization and machine learning.
>
>     - **Preimage-based initialization:** In black-box optimization, it is important to select initial data that broadly covers the search space [1]. Since coverage increases when the data distribution is closer to that of the search space and when the number of data points is larger, we designed Eq. 7 to measure the coverage.
>     - **Consistency regularization:** We encourage the regression model to predict identical predictions for soft prompts within the same preimage to impose the preimage structure as a prior. Such regularization is a well-established technique across various areas of machine learning.
>
>     Theoretical justification for PRESTO is a highly intriguing direction, and we also have a deep interest in exploring it further.
>
>     - Reference
>
>         [1] Jones, Donald R., Matthias Schonlau, and William J. Welch. "Efficient global optimization of expensive black-box functions." *Journal of Global optimization* (1998).
>
> - **[W3] Wall-clock time comparison**
>
>
>     |  | InstructZero [1] | INSTINCT [2] | ZOPO [3] | PRESTO (Ours) |
>     | --- | --- | --- | --- | --- |
>     | Preprocess (min.) | - | 2.02 $\pm$ 0.38 | 5.07 $\pm$ 0.48 | 6.81 $\pm$ 0.51 |
>     | Optimization (min.) | 11.17 $\pm$ 1.04 | 13.21 $\pm$ 1.79 | 9.53 $\pm$ 1.19 | 10.63 $\pm$ 1.36 |
>     | Average accuracy | 61.67 | 67.92 | 69.79 | **72.76** |
>
>     We conducted a wall-clock time comparison with baselines. During preprocessing, which is performed before the optimization process begins, PRESTO generates both LLM embeddings and instructions, whereas INSTINCT generates only the embeddings. Despite involving more components, PRESTO achieves a lower overall optimization time than INSTINCT. This is because PRESTO pre-generates instructions in batch during preprocessing, while INSTINCT queries the LLM at every optimization step. As shown above, PRESTO incurs only marginal preprocessing overhead, yet achieves superior optimization performance.
>
>     - Reference
>
>         [1] Chen, Lichang, et al. "Instructzero: Efficient instruction optimization for black-box large language models." ICML (2024).
>
>         [2] Lin, Xiaoqiang, et al. "Use your instinct: Instruction optimization for llms using neural bandits coupled with transformers." ICML (2024).
>
>         [3] Hu, Wenyang, et al. "Localized zeroth-order prompt optimization." NeurIPS (2024).
>
> - **[W4] Performance of PRESTO on complicated tasks**
>
>     PRESTO demonstrates strong performance on both simple and complex tasks. Interestingly, even when the task is more complex or challenging, the proposed method still boosts performance, while the baselines are NOT effective in optimizing prompts. For instance, on a math reasoning task, AQUA-RAT, which is a relatively challenging task, INSTINCT provides only a marginal improvement (0.78%) over hand-crafted instruction. However, our method achieves a significant performance gain of 3.54% over hand-crafted instruction. Although we cannot provide more results at the moment due to limited resources, we will include more challenging tasks in the final version.

---

> > ### Comment · Reviewer_cs11 · 2025-08-05
> >
> > Thank you for the detailed rebuttal. I appreciate the new wall-clock time comparison, which helps address my questions about overhead.
> >
> > I still believe that the work can be enhanced with deeper theoretical justification for its heuristic design choices. Furthermore, a **qualitative analysis** is still needed to explain why the method's instructions are more effective on complex tasks. This would provide valuable insight that **the *dry* accuracy metrics alone do not capture**.
> >
> > As mentioned in my initial review, I will maintain my positive rating given the paper's strong empirical contributions.

---

> > > ### Author Response · Authors · 2025-08-06
> > >
> > > We appreciate the thoughtful feedback. Qualitative analysis and theoretical justification are important directions, and we plan to incorporate qualitative results in the final version while leaving the full theoretical investigation to future work. We appreciate your feedback and will incorporate it into the final version.

---

### Official Review · Reviewer_BKWA · 2025-07-03

**Clarity:** 3
**Significance:** 3
**Originality:** 3
**Rating:** 4
**Confidence:** 3

**Summary:**

This paper introduces PRESTO, a novel framework for optimizing instructions for black-box large language models (LLMs) by leveraging the preimage structure—the many-to-one mapping from soft prompts to instructions produced by a white-box LLM. Unlike previous work, which treats this redundancy as a nuisance, PRESTO exploits it to improve sample efficiency and optimization performance. The framework consists of three main components: (1) score sharing across soft prompts in the same preimage, (2) preimage-based initialization to increase search space coverage, and (3) score consistency regularization for the score predictor. Experiments on 33 instruction optimization tasks show that PRESTO outperforms strong baselines.

**Questions:**

Ablation Studies:
The paper would be improved by including ablations isolating the impact of each PRESTO component (score sharing, initialization, regularization) to better understand their individual contributions.

**Ethical Concerns:**

["NO or VERY MINOR ethics concerns only"]

**Final Justification:**

The authors' response clarified several of my concerns, particularly regarding scalability, computational overhead, and the empirical contributions of each PRESTO component. The added analyses and ablations strengthen the paper’s technical grounding and confirm its effectiveness.

However, two core limitations remain: (1) the dependence on access to white-box LLMs, and (2) assumptions about the feasibility of constructing preimage structures in broader settings. These restrict general applicability beyond the evaluated scenarios.

Overall, I find the work well-executed and valuable within its scope. My score remains unchanged, reflecting my assessment of the work.

**Limitations:**

yes

**Quality:**

3

**Strengths And Weaknesses:**

## Strengths

- Novelty. The key insight—interpreting the many-to-one mapping from soft prompts to instructions as a source of prior knowledge rather than redundancy—is both original and well-motivated. This perspective enables PRESTO to achieve significant improvements in query efficiency and optimization performance

- Empirical Results. The method is evaluated on a broad set of 33 instruction optimization tasks (30 induction, 3 arithmetic reasoning), achieving state-of-the-art performance compared to strong baselines such as INSTINCT and ZOPO. PRESTO shows the effect of observing up to 14 times more scored data under the same query budget.
Clarity. The paper is well-organized, with clear motivation, background, and methodological exposition. The figures (e.g., distribution of preimage sizes) effectively illustrate the core concepts.

## Weaknesses

- Assumptions about Preimage Structure: PRESTO assumes that the preimage structure (i.e., which soft prompts map to which instructions) can be efficiently established by sampling a large set of soft prompts and mapping them through the white-box LLM. The scalability of this approach to very large prompt spaces or more complex instruction sets is not fully discussed.
- Dependence on White-Box LLMs: The framework relies on access to a white-box LLM to generate instructions from soft prompts and to extract embeddings for the score predictor. This may limit applicability in scenarios where only black-box LLMs are available, or where the white-box model's behavior diverges from the black-box model.
- Evaluation Scope: While the paper reports strong results on 33 tasks, it is unclear how PRESTO performs on tasks with fundamentally different instruction distributions or in settings where the preimage sizes are much smaller (i.e., less redundancy to exploit). Additional ablations or analysis on such edge cases would strengthen the empirical claims.
- Computational Overhead: The need to precompute and store large sets of soft prompts, instructions, and preimage mappings may incur significant computational and memory overhead, especially as the candidate set grows. The paper would benefit from a more detailed discussion of these practical considerations.

---

> ### Author Rebuttal · Authors · 2025-07-31
>
> - **[W1] Scalability analysis of preimage construction**
>
>
>     | # Soft prompt tokens | 3 | 5 | 10 | 50 | 100 |
>     | --- | --- | --- | --- | --- | --- |
>     | Preimage construction (min.) | 6.72 $\pm$ 0.39 | 6.87 $\pm$ 0.44 | 7.08 $\pm$ 0.45 | 8.66 $\pm$ 0.62 | 10.52 $\pm$ 0.70 |
>
>     Thank you for the suggestion. We provide a scalability analysis of preimage construction with respect to the size of the soft prompt space, which is defined as (number of tokens $\times$ dimension). Since the dimension is fixed (it depends on the white-box LLM), we focus on the number of soft prompt tokens: 3, 5, 10, 50, and 100. The table above shows that the proposed method has good scalability. With a large number of soft prompts (50 and 100), the preimage construction remains computationally feasible. In our experiments, we used 3 to 10 soft prompt tokens.
>
> - **[W2] Availability of and dependency on white-box LLM**
>
>     Scenarios where only black-box LLMs are available are relatively uncommon, since various open-source white-box LLMs are already publicly available. Several prior studies have leveraged white-box LLMs to facilitate instruction optimization for black-box LLMs [1,2,3]. Additionally, as already shown in Tables 4 and 5 of the supplementary material, PRESTO consistently achieves strong performance across diverse white-box and black-box LLM combinations.
>
>     - Reference
>
>         [1] Chen, Lichang, et al. "Instructzero: Efficient instruction optimization for black-box large language models." ICML (2024).
>
>         [2] Lin, Xiaoqiang, et al. "Use your instinct: Instruction optimization for llms using neural bandits coupled with transformers." ICML (2024).
>
>         [3] Hu, Wenyang, et al. "Localized zeroth-order prompt optimization." NeurIPS (2024).
>
> - **[W3] Analysis of the effect of preimage size**
>
>
>     | Preimage size (%) | \- (Vanilla) | 1 | 10 | 50 | 100 (Current) |
>     | --- | --- | --- | --- | --- | --- |
>     | Average accuracy | 51.91 | 59.95 | 60.67 | 61.89 | 62.91  |
>
>     As suggested, we analyzed the effect of preimage size by varying the proportion of soft prompts included in each preimage (1%, 10%, 50%, and 100%). We also included experimental results for the vanilla model, which does not utilize the preimage structure. The results show that larger preimage sizes lead to higher average accuracy, indicating that richer information in the preimage facilitates more successful optimization. This highlights the critical role of the preimage structure in instruction optimization.
>
> - **[W4] Computational analysis of preimage construction**
>
>
>     | Candidate set size | 1k | 5k | 10k (Current) | 20k | 30k |
>     | --- | --- | --- | --- | --- | --- |
>     | Preimage construction (min.) | 0.66 $\pm$ 0.04 | 3.31 $\pm$ 0.20 | 6.72 $\pm$ 0.39 | 13.36 $\pm$ 0.80 | 19.82 $\pm$ 1.02 |
>     | Memory (MB) | 47.47 | 237.46 | 474.96 | 950.10 | 1425.63 |
>
>     We provide an efficiency analysis of the preimage construction process with respect to the candidate set size. The results show that construction time and total memory usage increase approximately linearly with the size of the candidate set, while the overall cost remains modest.
>
> - **[Q1] Additional ablation studies**
>
>
>     | Model | Vanilla | + Init | + Reg | + SS |
>     | --- | --- | --- | --- | --- |
>     | Avg. acc. | 51.91 | 52.41 (+0.5) | 53.49 (+1.58)  | 59.57 (+7.66) |
>
>     We have conducted additional ablation studies that apply score consistency regularization (Reg) and preimage-based initialization (Init) to the vanilla model. Preimage-based initialization shows relatively smaller performance gain since this method assumes the presence of score sharing. As shown in Table 3 of the main paper, adding preimage-based initialization to score sharing results in a further performance improvement (+2.25).

---

> ### Comment · Reviewer_BKWA · 2025-08-06
> **Response to authors**
>
> Thank you for the detailed response. The additional analyses on scalability, computational cost, and preimage size are helpful and clarify several concerns I raised. The ablation study on PRESTO's components also strengthens the empirical grounding of the method.
>
> That being said, the core limitations regarding dependency on a white-box LLM and the assumptions about preimage structure remain significant for broader applicability. While the response addresses these points to an extent, my current scores accurately reflect my assessment of the work.
>
> I believe the submission is technically solid and well-executed within its scope, but the limitations noted constrain its generality. My current score accurately reflects my assessment of the work, and I will be keeping it unchanged.

---

> > ### Author Response · Authors · 2025-08-06
> >
> > Thank you for your thoughtful comments.
> >
> > As we noted in the rebuttal, we believe that the assumption of access to white-box LLMs is consistent with the majority of existing literature. Additionally, we have included empirical evidence in Figure 2 of the supplementary material showing that a meaningful preimage structure exists across various white-box LLMs, Qwen2.5-7B-Instruct and Mistral-7B-Instruct-0.3v. These experiments demonstrate that various white-box LLMs possess sufficient preimage structures to support our method.
> >
> > We appreciate your feedback and will incorporate it into the final version.

---

### Official Review · Reviewer_uyym · 2025-07-03

**Clarity:** 3
**Significance:** 3
**Originality:** 3
**Rating:** 5
**Confidence:** 4

**Summary:**

This paper tackles the problem of automatically optimizing instructions for black-box LLMs by leveraging an auxiliary white-box LLM. Prior work generates soft prompts that are fed into a white-box model to propose candidate instructions, but often different soft prompts map to the same instruction. Rather than regarding this redundancy as wasteful, the authors introduce PRESTO, which exploits the preimage structure to amplify the effective data and guide search. There are 3 major components in the proposed method: (1) score sharing, which shares the evaluation score with all soft prompts in a preimage; (2) preimage-based initialization, which selects initial data points that maximize search space coverage using preimage information; and (3) score consistency regularization, which enforces prediction consistency within each preimage. Empirically, PRESTO uses the same query budget but effectively learns from 14× more “virtual” data points, yielding superior performance on 33 instruction-induction tasks and three arithmetic reasoning benchmarks, outperforming strong baselines.

**Questions:**

How will the cost of MMD scale with the candidate set size?

**Ethical Concerns:**

["NO or VERY MINOR ethics concerns only"]

**Limitations:**

Yes

**Quality:**

3

**Strengths And Weaknesses:**

Strength:

1. The idea of reinterpreting the many-to-one mapping as useful prior knowledge to enhance the optimization performance is novel.

2. The experimental results on a variety of datasets show the effectiveness of the proposed method over the compared baselines.

Weakness:

In general I think the paper is good. However, I have one concern - though the method is designed to improve the data efficiency, I am curious about the overall computational efficiency of the proposed method. For example, how will the cost of MMD scale with the candidate set size? A more detailed comparison of the algorithm efficiency with the baseline method is desired.

---

> ### Author Rebuttal · Authors · 2025-07-31
>
> - **[W1, Q1] Computational Analysis of MMD**
>
>
>     | Candidate set size | 1k | 5k | 10k (Current) | 20k | 30k |
>     | --- | --- | --- | --- | --- | --- |
>     | MMD computation time (sec.) | 7.18 $\pm$ 1.39 | 11.13 $\pm$ 1.77 | 27.67 $\pm$ 3.75  | 79.92 $\pm$ 5.32 | 94.31 $\pm$ 8.83 |
>
>     Thank you for the suggestion. We conducted a computational analysis of the MMD with respect to the size of the candidate set (1k, 5k, 10k, 20k, 30k). As expected, the computation time increases with the size of the candidate set. Notably, even the largest setting (30k) remains computationally feasible, taking approximately 1 minute and 30 seconds.
>
> - **[W1] Efficiency Comparison with Baselines**
>
>
>     |  | InstructZero [1] | INSTINCT [2] | ZOPO [3] | PRESTO (Ours) |
>     | --- | --- | --- | --- | --- |
>     | Preprocess (min.) | - | 2.02 $\pm$ 0.38 | 5.07 $\pm$ 0.48 | 6.81 $\pm$ 0.51 |
>     | Optimization (min.) | 11.17 $\pm$ 1.04 | 13.21 $\pm$ 1.79 | 9.53 $\pm$ 1.19 | 10.63 $\pm$ 1.36 |
>     | Average accuracy | 61.67 | 67.92 | 69.79 | **72.76** |
>
>     We conducted a wall-clock time comparison with baselines. During preprocessing, which is performed before the optimization process begins, PRESTO generates both LLM embeddings and instructions of candidate soft prompts, whereas INSTINCT generates only the embeddings. Despite involving more components, PRESTO achieves a lower overall optimization time than INSTINCT. This is because PRESTO pre-generates candidate instructions in batch during preprocessing, while INSTINCT queries the LLM at every optimization step. As shown above, PRESTO incurs only marginal preprocessing overhead, yet achieves superior optimization performance.
>
>     - Reference
>
>         [1] Chen, Lichang, et al. "Instructzero: Efficient instruction optimization for black-box large language models." ICML (2024).
>
>         [2] Lin, Xiaoqiang, et al. "Use your instinct: Instruction optimization for llms using neural bandits coupled with transformers." ICML (2024).
>
>         [3] Hu, Wenyang, et al. "Localized zeroth-order prompt optimization." NeurIPS (2024).

---

> ### Comment · Reviewer_uyym · 2025-08-08
>
> Thanks for the response. I think the paper is in general solid. So I'll keep my score unchanged.

---

### Decision · Program_Chairs · 2025-09-17

**Decision:**

Accept (poster)

**Comment:**

This paper introduces PRESTO. The key contribution is to reinterpret the many-to-one mapping from soft prompts to discrete instructions, a phenomenon previously seen as a redundancy, as a useful structure called a "preimage". The paper's primary strengths, as noted by the reviewers, include the  novelty of its core insight and its strong, extensive empirical results across 33 tasks and multiple LLMs. Weaknesses raised during the review process include a  dependency on access to white-box LLMs, a  lack of formal theoretical justification for some heuristic design choices , and initial concerns about computational overhead (the last of which were effectively addressed during rebuttal).

During the rebuttal period, the authors effectively addressed most of the reviewers' concerns. They provided a detailed wall-clock time comparison and scalability analysis, demonstrating that the computational overhead is manageable and competitive with baselines. They also provided requested ablation studies that clarified the contribution of each of PRESTO's components. Overall, a majority of reviewers acknowledge the novelty of the work, and there is consensus on its strong empirical results, both in terms of the margin of gain and the extensiveness of the evaluation. The rebuttal cleared most of the issues raised. The remaining weaknesses concern the lack of deep theoretical justification (which is a valid point but is arguably what the entire field is lacking) and the reliance on white-box LLMs (again, this is a setup shared across all methods based on soft prompt tuning). All reviewers ultimately recommended acceptance. Hence, the AC recommends acceptance of this paper.